# KD-MRI: A knowledge distillation framework for image reconstruction and image restoration in MRI workflow

**Balamurali Murugesan** [1,2]          BALAMURALI@HTIC.IITM.AC.IN
[1] *Indian Institute of Technology Madras (IITM), India*
[2] *Healthcare Technology Innovation Centre (HTIC), IITM, India*

**Sricharan Vijayarangan** [2*]          SRICHARANV@HTIC.IITM.AC.IN
**Kaushik Sarveswaran** [2*†]          KAUSHIK3497@YAHOO.CO.IN
**Keerthi Ram** [2]                KEERTHI@HTIC.IITM.AC.IN
**Mohanasankar Sivaprakasam** [1,2]       MOHAN@EE.IITM.AC.IN

## Abstract

Deep learning networks are being developed in every stage of the MRI workflow and have provided state-of-the-art results. However, this has come at the cost of increased computation requirement and storage. Hence, replacing the networks with compact models at various stages in the MRI workflow can significantly reduce the required storage space and provide considerable speedup. In computer vision, knowledge distillation is a commonly used method for model compression. In our work, we propose a knowledge distillation (KD) framework for the image to image problems in the MRI workflow in order to develop compact, low-parameter models without a significant drop in performance. We propose a combination of the attention-based feature distillation method and imitation loss and demonstrate its effectiveness on the popular MRI reconstruction architecture, DC-CNN. We conduct extensive experiments using Cardiac, Brain, and Knee MRI datasets for 4x, 5x and 8x accelerations. We observed that the student network trained with the assistance of the teacher using our proposed KD framework provided significant improvement over the student network trained without assistance across all the datasets and acceleration factors. Specifically, for the Knee dataset, the student network achieves 65% parameter reduction, 2x faster CPU running time, and 1.5x faster GPU running time compared to the teacher. Furthermore, we compare our attention-based feature distillation method with other feature distillation methods. We also conduct an ablative study to understand the significance of attention-based distillation and imitation loss. We also extend our KD framework for MRI super-resolution and show encouraging results.

**Keywords:** MRI workflow, Model compression, Knowledge distillation, MRI reconstruction, MRI super resolution.

## 1. Introduction

Magnetic Resonance Imaging (MRI) workflow consists of image acquisition, reconstruction, restoration, registration and analysis (Lundervold and Lundervold, 2019). In every stage of the MRI pipeline, deep learning networks have shown encouraging results and are being

---

* Equal contribution.
† Work done while interning at HTIC.

integrated into the medical workflow (Thaler and Menkovski, 2019). This integration demands larger storage and compute power as the improved performance of deep networks come at the cost of computation and storage. Consequently, hospitals which are already burdened with storing large medical records will now have to allocate additional storage for the deep learning models. Furthermore, with the advent of patient-specific care (Vivanti et al., 2018) and federated learning (Konečný et al., 2016), the need for storage and compute power will continue to increase.

Deep networks are task specific, separate networks are required for image segmentation, image reconstruction, image super-resolution, object detection, etc.. Thereby, for the different tasks in each stage in MRI workflow, individual networks are developed. In addition to the task specific nature of deep learning, they are also dataset specific. Deep learning networks developed for a particular task using a certain dataset might perform poorly on a new dataset from a different distribution. In MRI, dataset is decided by the anatomical study (brain, cardiac, knee) and its respective contrast (T1, T2). So, for every task in MRI workflow, specific deep networks are to be developed with respect to a particular dataset. Furthermore, for tasks like reconstruction, apart from the choice of dataset, the degradation caused to the input image is varied through different acceleration factor (2x, 4x, 8x) and undersampling mask (cartesian, gaussian) causing a change in distribution. Due to the plethora of configurations (task, dataset, type of degradation) to be considered in an MRI workflow, the cost of deploying existing state-of-the-art deep networks at each stage accumulates to an exponential increase in memory and computation. Hence, there is a pressing need for memory-efficient model development.

Model compression is an actively pursued area of research over the last few years with the goal of deploying state-of-the-art deep networks in low-power and resource limited devices without significant drop in accuracy (Cheng et al., 2017). Parameter pruning, low-rank factorization and weight quantization are some of the proposed methods to compress the size of deep networks. However, these methods may require dedicated hardware or software customization for practical implementation. A promising method to obtain compact models with ease of deployment is Knowledge distillation (KD) (Hinton et al., 2015). In KD, the student model (memory efficient network) learns from the powerful teacher model (state-of-the-art network) to improve the student's accuracy which drops due to parameter reduction. In computer vision, KD has been widely developed for image classification tasks (Romero et al., 2015) (Mirzadeh et al., 2019). Recently, some of the works have focused on applying KD to image segmentation (Liu et al., 2019) and object detection (Chen et al., 2017) tasks. These works can be adapted to the MRI analysis stage. In our work, we propose to compress the deep learning models in reconstruction and restoration stage through our novel KD framework. Thereby, the entire MRI workflow can be implemented with efficient storage and computation with significant speed-up. We primarily use MRI reconstruction to demonstrate the effectiveness of our proposed framework. We also extend our framework for MRI super-resolution and obtained encouraging results which are presented in Appendix A. In summary, the following are our contributions:

- We propose an end-to-end trainable framework for learning compact MRI reconstruction networks through knowledge distillation (KD). To the best of our knowledge, this is the first application of KD for the MRI reconstruction problem.

- For MRI reconstruction and restoration, we propose an attention-based feature distillation method, which helps the student learn the intermediate representation of the teacher. We also propose combining it with imitation loss function based KD, which acts as a regularizer to the reconstruction loss.

- We demonstrate the effectiveness of our approach using deep cascade of convolutional neural network (DC-CNN). We use DC-CNN with five cascades and three convolution layers as student (S-DC-CNN), five cascades and five convolution layers as teacher (T-DC-CNN). We perform extensive experiments using Cardiac, Brain, and Knee MRI dataset for 4x, 5x, and 8x accelerations. We show that S-DC-CNN trained using our KD method showed consistent improvement of PSNR and SSIM over the S-DC-CNN trained without assistance of the teacher across all datasets and acceleration factors. Considering Knee image reconstruction, S-DC-CNN gives 65% parameter reduction, 2x faster CPU running time, and 1.5x faster GPU running time compared to T-DC-CNN.

- We compare our attention based feature distillation method against common feature distillation methods. We observed that our method provides lower validation error and is thus better in transferring teacher's knowledge to student. We also conduct an ablative study to understand the significance of our attention-based feature distillation and imitation loss. We found that attention transfer is the key to KD.

## 2. Brief Literature Review

### 2.1. MRI reconstruction

MRI reconstruction is the process of transforming the acquired Fourier space (k-space) data to image domain. Since MRI is a slow acquisition modality, only samples (under sampled) of k-space data are acquired to obtain the reconstruction. However, this reconstruction suffers from aliasing artifacts. Currently, deep learning networks are developed to de-alias the artifact and provide reconstruction equivalent of sampling the entire (fully sampled) k-space. (Wang et al., 2016) proposed a basic convolution neural network (CNN) to learn the representation between under sampled (US) and fully sampled (FS) image. Later, (Lee et al., 2017) introduced residual learning which showed that learning the aliasing artifacts is easier and better than learning the FS image. (Zhu et al., 2018) proposed AUTOMAP, a fully connected network to operate on the k-space domain to learn the mapping between US k-space and FS image. (Hyun et al., 2018) proposed to use U-Net with data consistency (DC) block to retain the known frequency components in predicted FS image. (Schlemper et al., 2017) introduced DC-CNN, a deep cascade network with each cascade containing CNN and DC blocks. (Sun et al., 2019) replaced CNN in DC-CNN with U-Net.

### 2.2. Knowledge distillation

(Hinton et al., 2015) introduced the concept of KD in deep neural networks for model compression. In their work, they proposed a student-teacher paradigm where the student, a lesser parameter network, obtains the knowledge from the teacher by learning the class distributions via the softmax layer. (Romero et al., 2015) proposed hint training, (Yim

et al., 2017) introduced Flow of Solution Procedure (FSP), (Komodakis and Zagoruyko, 2017) developed attention mechanism which enables student to learn the intermediate representations of teacher. Unlike previous works, (Chen et al., 2017) proposed a knowledge transfer procedure for regression based on teacher bounded loss. Recently, (Saputra et al., 2019) proposed various ways of blending the loss of the student with respect to the ground truth and the teacher.

## 3. Methodology

### 3.1. MRI reconstruction problem formulation

Let $\mathbf{x} \in C^N$ represent a column stacked vector of complex valued MR image with dimension $\sqrt{N} \times \sqrt{N}$. Let $\mathbf{y} \in C^M$ represent the undersampled k-space measurements. By definition, $\mathbf{y} = \mathbf{F}_u \mathbf{x}$, where $\mathbf{F}_u \in C^{M \times N}$ is an undersampled fourier encoding matrix. Our problem is to reconstruct $\mathbf{x}$ from $\mathbf{y}$. This linear inversion $\mathbf{x}_u = \mathbf{F}_u^H \mathbf{y}$ is fundamentally ill-posed and generates an aliased image due to violation of Nyquist-Shannon sampling theorem. The deep learning formulation to obtain $\mathbf{x}$ is given by:

$$\min_{x,\theta} \quad ||\mathbf{x} - f_{cnn}(\mathbf{x}_u|\theta)||_2^2 + \lambda ||\mathbf{F}_u \mathbf{x} - \mathbf{y}||_2^2 \tag{1}$$

where $f_{cnn}$ is the deep network parameterised by $\theta$, which learns the mapping between $\mathbf{x}_u$ and $\mathbf{x}$. To provide data consistency for the network's output, the following data fidelity procedure is followed:

$$\hat{\mathbf{x}}_{rec} = \begin{cases} \hat{\mathbf{x}}_{cnn}(k) & k \notin \Omega \\ \frac{\hat{\mathbf{x}}_{cnn}(k) + \lambda \hat{\mathbf{x}}_u(k)}{1+\lambda} & k \in \Omega \end{cases} \tag{2}$$

where $\hat{\mathbf{x}}_{cnn} = \mathbf{F} x_{cnn}, x_{cnn} = f_{cnn}(\mathbf{x}_u|\theta)$, $\hat{\mathbf{x}}_u = \mathbf{F}\mathbf{x}_u$, $\mathbf{x}_{rec} = \mathbf{F}^{-1}\hat{\mathbf{x}}_{rec}$, $\Omega$ is an index set indicating which k space measurements have been sampled.

### 3.2. Proposed knowledge distillation framework

KD is the process of transferring the knowledge of a large teacher network to a small student network. The main idea of KD is to achieve parameter reduction without significant drop in performance. In our work, we design KD methods for MRI reconstruction and demonstrate its efficacy by applying it to the commonly used DC-CNN network. This choice was made considering the following factors: 1) Simple design involving fully convolutional layers, 2) Extensibility to other CNN based MRI-Reconstruction architectures ((Souza et al., 2019),(Wu et al., 2018),(Sun et al., 2018)). Figure 1 depicts the overview of DC-CNN using KD.

**Deep cascade of convolutional neural network (DC-CNN)** DC-CNN (Schlemper et al., 2017) consists of a cascade of convolution layers and a data consistency (DC) layer. The number of cascades in DC-CNN is given by $n_c$. A single cascade has $n_d$ convolution layers and 1 DC layer. The kernel size for each convolution layer is $3 \times 3$ with stride and padding set to 1. The initial convolution layer takes 1 channel (real) as input (US image) and gives 32 feature maps while the final convolution layer takes 32 feature maps as input and gives 1 channel as output (FS image). The number of input and output feature maps for the other convolution layers is 32. ReLU is used to introduce non-linearity between convolution

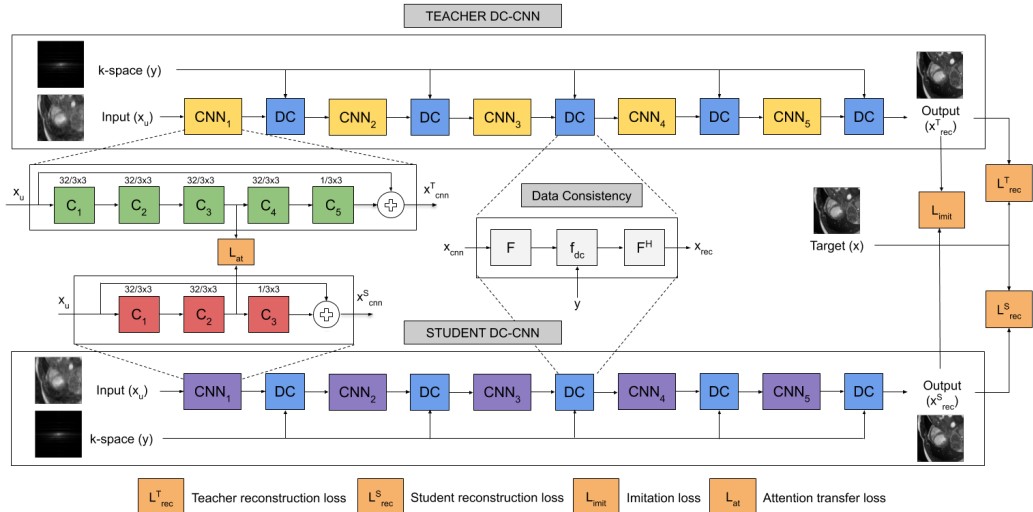

Figure 1: Teacher DC-CNN: Five cascades with each cascade having five convolution layers. Student DC-CNN: Five cascades with each cascade having three convolution layers. Attention transfer and imitation loss helps in teacher-student knowledge transfer. Attention transfer loss is obtained between the output of third and second convolution layer of each cascade in Teacher and Student DC-CNN. Imitation loss is obtained between the outputs of Teacher and Student DC-CNN.

layers. DC layer fill the predicted k-space with known values to provide consistency in Fourier domain. The cascade has a residual connection which sums the output of the cascade with its input.

**Teacher DC-CNN**: DC-CNN with five cascades ($n_c = 5$) and five convolution layers ($n_d = 5$) is chosen as teacher. Let $f_{cnn}^T$ parametrized by $\theta^T$ be the teacher DC-CNN. Then, reconstruction $\mathbf{x}_{rec}^T$ from teacher is given by: $\mathbf{x}_{rec}^T = f_{cnn}^T(\mathbf{x}_u | \theta^T)$.

**Student DC-CNN**: DC-CNN with five cascades ($n_c = 5$) and three convolution layers ($n_d = 3$) is chosen as student. Let $f_{cnn}^S$ parametrized by $\theta^S$ be the student DC-CNN. Then, the reconstruction $\mathbf{x}_{rec}^S$ from student is given by: $\mathbf{x}_{rec}^S = f_{cnn}^S(\mathbf{x}_u | \theta^S)$.

### 3.2.1. ATTENTION-BASED FEATURE DISTILLATION

(Komodakis and Zagoruyko, 2017) used attention maps as a feature distiller and showed improvement in classification performance of student networks. For classification tasks, the attention maps provide the significance of each activation in the feature map w.r.t the input. However, in the case of image-to-image regression problems, the attention maps would provide an intermediate image reflecting the final reconstructed output. Hence, this would provide the most direct form of teacher supervision for MRI Reconstruction. Thus, the goal is to make the student network mimic the attention map of teacher network allowing it to learn better intermediate representations.

Let the feature maps after activation be denoted as $A \in R^{C \times H \times W}$, where C is the number of channels and $H \times W$ is the spatial dimension. The attention map of the features is given by $F_{sum}(A) = \sum_{i=1}^{C} |A_i|^2$. To obtain effective information distillation, we adapt the following attention transfer loss:

$$L_{AT} = \sum_{j \in I} ||\frac{Q_S^j}{||Q_S^j||_2} - \frac{Q_T^j}{||Q_T^j||_2}||_2 \tag{3}$$

where $Q_S^j = vec(F_{sum}(A_S^j))$ and $Q_T^j = vec(F_{sum}(A_T^j))$ represent the $j$-th pair of student and teacher attention maps in vectorized form, $I$ denote the set of teacher-student convolution layers which is selected for attention transfer. In our case, the convolution layer at the center of each cascade from teacher and student DC-CNN form the set $I$. This choice of distillation position was made after trying different combinations which is reported in Appendix B.

### 3.2.2. Imitation loss

(Saputra et al., 2019) proposed to use the imitation loss as an additional constraint along with the student loss and showed performance improvement in student networks. We incorporate this loss in our MRI reconstruction problem as it can act as a regularizer to the student reconstruction loss. As this constraint is enforced along with the regular reconstruction loss, there is no additional overhead in terms of training time unlike the attention transfer. Herein, we propose a total reconstruction loss for student as follows:

$$L_{total}^S = \alpha L_{rec}^S + (1 - \alpha)L_{imit} \tag{4}$$

where $L_{rec}^S = ||x - x_{rec}^S||$ is the loss between student prediction and target, $L_{imit} = ||x_{rec}^T - x_{rec}^S||$ is the imitation loss between teacher and student prediction

### 3.2.3. Train procedure

---
**Algorithm 1:** Knowledge transfer procedure

---
- Step1: Train the teacher DC-CNN $f_{cnn}^T$ weights $\theta^T$ using teacher reconstruction loss $L_{rec}^T = ||x - x_{rec}^T||$;

- Step2: Train the student DC-CNN $f_{cnn}^S$ weights $\theta^S$ using attention transfer loss $L_{AT} = ||Q_T - Q_S||$ between teacher and student;

- Step3: Load the weights $\theta^S$ from Step2 and re-train $f_{cnn}^S$ weights $\theta^S$ using student reconstruction and imitation loss $L_{total}^S = \alpha||x - x_{rec}^S|| + (1 - \alpha)||x_{rec}^T - x_{rec}^S||$;

---

## 4. Experiments and Results

### 4.1. Dataset Description, Evaluation metrics and Implementation details

**Dataset Description:** 1) **Cardiac MRI dataset**: Automated Cardiac Diagnosis Challenge (ACDC) (Bernard et al., 2018) consists of 150 and 50 patient records for training and

Table 1: Quantitative comparison between Zero-filled (ZF)(US Image), Teacher (T-DC-CNN), Student (S-DC-CNN) and our proposed model (S-KD-DC-CNN) across PSNR and SSIM metrics for ACDC, MRBrainS and Knee MRI datasets. Red indicates best and blue indicates second best performance.

| | | 4x | | 5x | | 8x | |
|---|---|---|---|---|---|---|---|
| | | PSNR | SSIM | PSNR | SSIM | PSNR | SSIM |
| Cardiac | ZF | 24.27 ± 3.10 | 0.6996 ± 0.08 | 23.82 ± 3.11 | 0.6742 ± 0.08 | 22.83 ± 3.11 | 0.6344 ± 0.09 |
| | Teacher | 32.51 ± 3.23 | 0.9157 ± 0.04 | 31.49 ± 3.32 | 0.9002 ± 0.04 | 28.43 ± 3.13 | 0.8335 ± 0.06 |
| | Student | 31.92 ± 3.17 | 0.9053 ± 0.04 | 30.79 ± 3.24 | 0.8863 ± 0.05 | 27.87 ± 3.11 | 0.8156 ± 0.07 |
| | Ours | 32.07 ± 3.21 | 0.9084 ± 0.04 | 31.01 ± 3.27 | 0.8913 ± 0.04 | 28.11 ± 3.17 | 0.8236 ± 0.07 |
| Brain | ZF | 31.38 ± 1.02 | 0.6651 ± 0.02 | 29.93 ± 0.80 | 0.6304 ± 0.02 | 29.37 ± 0.98 | 0.6065 ± 0.03 |
| | Teacher | 40.03 ± 2.00 | 0.9781 ± 0.00 | 39.03 ± 1.28 | 0.971 ± 0.00 | 35.04 ± 1.38 | 0.9374 ± 0.01 |
| | Student | 39.36 ± 1.82 | 0.9753 ± 0.00 | 38.58 ± 1.28 | 0.9674 ± 0.00 | 34.39 ± 1.26 | 0.9281 ± 0.01 |
| | Ours | 39.8 ± 1.89 | 0.977 ± 0.00 | 38.78 ± 1.24 | 0.9688 ± 0.00 | 34.83 ± 1.35 | 0.9337 ± 0.01 |
| Knee | ZF | 29.66 ± 3.86 | 0.8066 ± 0.08 | 29.2 ± 3.87 | 0.8007 ± 0.08 | 28.71 ± 3.88 | 0.7985 ± 0.08 |
| | Teacher | 37.15 ± 3.55 | 0.9436 ± 0.03 | 35.16 ± 3.46 | 0.9231 ± 0.03 | 32.53 ± 3.49 | 0.8887 ± 0.05 |
| | Student | 36.37 ± 3.53 | 0.9367 ± 0.03 | 34.37 ± 3.47 | 0.9144 ± 0.04 | 31.92 ± 3.58 | 0.8804 ± 0.05 |
| | Ours | 36.7 ± 3.52 | 0.9392 ± 0.03 | 34.71 ± 3.44 | 0.9181 ± 0.04 | 32.32 ± 3.57 | 0.8867 ± 0.05 |

validation respectively. We extracted the 2D slices and cropped to 150×150. These amount to 1841 and 1076 for train and validation. 2) **Brain MRI dataset**: MRBrainS dataset (Mendrik et al., 2015) contains T1, T1-IR and T2-FLAIR volumes for 7 subjects. We use T1 MRI with size 240×240. For training and validation, 5 subjects with 240 slices and 2 subjects with 96 slices are used. 3) **Knee MRI dataset**: The dataset used by (Kerstin et al.) has coronal proton density knee volumes for 20 subjects acquired using 15-element knee coil. Each slice in the volume with dimension $640 \times 368$ has 15 channels and its respective sensitivity maps. The multi-channel slices are converted to single channel through root sum of squares. 10 subjects (200 slices) and other 10 (200 slices) are used for train and validation. US k-space and US images are retrospectively obtained using fixed cartesian undersampling masks for 4x, 5x and 8x acceleration factors. In ACDC and MRBrainS datasets, the undersampling masks sample ten while in the knee MRI dataset, thirty lowest spatial frequencies are sampled. The remaining frequencies follow a zero-mean Gaussian distribution. The undersampling masks can be found in Appendix C.

**Evaluation metrics**: Peak Signal-to-Noise Ratio (PSNR) and Structural Similarity Index (SSIM) metrics are used to evaluate the reconstruction quality. Wilcoxon signed-rank test with an alpha of 0.05 is used to assess statistical significance.

**Implementation Details** Models are implemented in PyTorch(0.4.0) [1]. $\alpha$ is set empirically to 0.5. For every step mentioned in Algorithm 1, models are trained for 150 epochs using the Adam optimizer, with a learning rate of 1e-4.

### 4.2. Results and discussion

#### 4.2.1. Quantitative and qualitative comparison

We compare S-KD-DC-CNN (Student DC-CNN trained using our KD procedure) with S-DC-CNN (Student DC-CNN trained without assistance of teacher) and T-DC-CNN (Teacher DC-CNN) for cardiac, brain and knee across 4x, 5x and 8x accelerations. Table 1 provides the quantitative comparison of the above methods. From the table, it can be observed that S-KD-DC-CNN provides significantly better performance than S-DC-CNN (Difference of S-

---

1. Code available at https://github.com/Bala93/KD-MRI

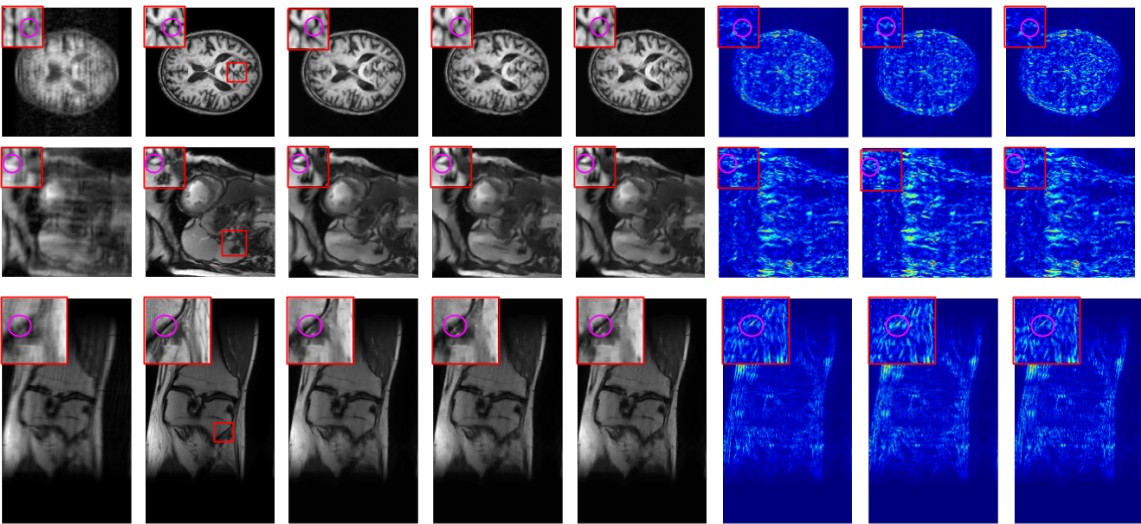

Figure 2: From Left to Right: Zero-filled, Target, Teacher (T-DC-CNN), Student (S-DC-CNN), Ours (S-KD-DC-CNN)), Teacher Residue, Student Residue, KD Residue. From Top to Bottom: MRBrainS, ACDC, Knee MRI. All the images are displayed for an acceleration factor of 5x. Upon examination, in addition to lower reconstruction errors the distilled model is able to retain finer structures better when compared to the student.

KD-DC-CNN and S-DC-CNN for PSNR and SSIM are statistically significant ($p < 0.05$)). This bridges the performance gap between teacher and student. Qualitative comparison of the methods are shown in Figure 2. In the Figure, it can be clearly seen that reconstructions obtained using S-KD-DC-CNN is closer to T-DC-CNN while S-DC-CNN has relatively higher information loss. The performance shown by S-KD-DC-CNN is due to the combination of Attention Transfer (AT) and imitation loss which help in reconstructing fine structures. The imitation loss is expected to behave as a regularizer to the student reconstruction loss while the AT assists the student to learn the intermediate representations of the teacher. We verified the same by obtaining residue between attention maps of T-DC-CNN and S-DC-CNN and compared it with the residue between attention maps of T-DC-CNN and S-KD-DC-CNN. We found that attention map of S-KD-DC-CNN is closer to T-DC-CNN compared to S-DC-CNN. Thus, learning these representations provide pre-trained weights which helps in more optimized training. Qualitative and quantitative comparison of the residues for each cascade along with the observations are reported in Appendix D.

### 4.2.2. PARAMETER COUNT AND RUNNING TIME

We calculate the parameter count and running time of T-DC-CNN and S-DC-CNN for Knee 4x acceleration to understand the effect of model compression. The parameter count of T-DC-CNN and S-DC-CNN are 141K and 49K respectively. The CPU running time for single

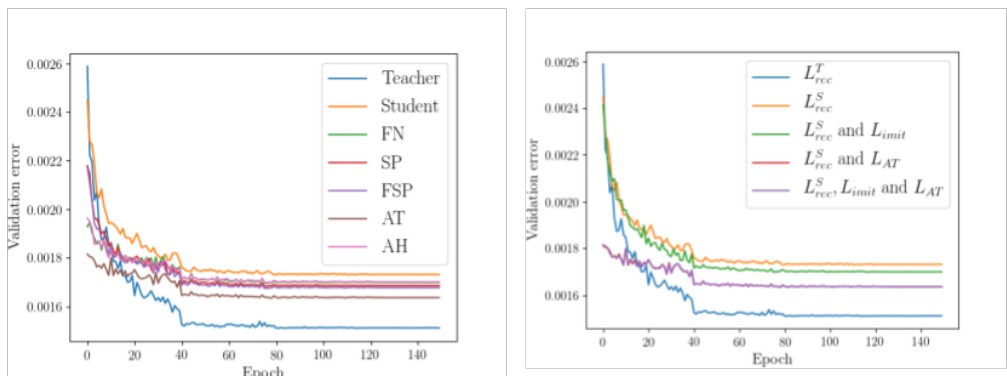

Figure 3: Left(a): Reconstruction loss of various feature distillation methods on the validation set. T-DC-CNN (Teacher), S-DC-CNN (Student), S-FN-DC-CNN (FN), S-FSP-DC-CNN (FSP), S-SP-DC-CNN (SP), S-AH-DC-CNN (AH) and S-AT-DC-CNN (Ours). Right(b): Ablation study of attention transfer and imitation loss functions.

image reconstruction for T-DC-CNN and S-DC-CNN are 568 ms and 294 ms respectively. The GPU running time for single image reconstruction of T-DC-CNN and S-DC-CNN are 24ms and 16ms respectively. As the number of parameters in S-KD-DC-CNN is equivalent to that of S-DC-CNN, we can state the following: 1) S-KD-DC-CNN gives 65% parameter reduction as compared to T-DC-CNN, 2) The CPU running time of S-KD-DC-CNN is nearly 2 times lower than that of T-DC-CNN, 3) The GPU running time of S-KD-DC-CNN is nearly 1.5 times lower than that of T-DC-CNN.

### 4.2.3. Comparison of feature distillation methods

We draw comparisons of our Attention Transfer method (AT) to other feature distillation methods, namely; FitNets (FN) (Romero et al., 2015), Flow of Solutions Procedure (FSP) (Yim et al., 2017), Similarity Preserving KD (SP) (Tung and Mori, 2019) and Attentive Hint (AH) (Saputra et al., 2019), for cardiac 8x acceleration. In this experiment, we pre-train the student network using weights obtained from feature distillation methods. During the fine tuning stage, we only consider the student reconstruction loss ignoring the imitation loss (by setting $\alpha = 1$ in Eq. 4). In FN, the student is expected to learn the entire feature map of the teacher. In the case of FSP, the student is entasked with mimicking the teacher in terms of the flow between the feature maps of the first and the penultimate layer. In SP, the student learns to mimic the similarity map of the intermediate layers of the teacher. In AH, teacher supervision is provided in a weighted fashion based on the reconstruction quality of the teacher. In AT, the student is expected to produce a sum of the feature maps in a fashion which is similar to that of the teacher. Figure 3a depicts the validation error loss for networks T-DC-CNN, S-DC-CNN, S-FN-DC-CNN, S-FSP-DC-CNN, S-SP-DC-CNN, S-AH-DC-CNN and S-AT-DC-CNN. The validation loss obtained using network S-AT-DC-CNN is lesser when compared to other methods and is thus, closer to the teacher

loss. This empirically demonstrates that AT is better at transferring the knowledge of the teacher to the student. The quantitative comparison of AT with other feature distillation methods can be found in Appendix E.

### 4.2.4. Ablative study of attention transfer and imitation loss

We conduct an ablative study to understand the effect of attention transfer and imitation loss for cardiac 8x acceleration. Figure 3b presents a validation error plot comparing S-DC-CNN trained using different combination of loss functions. From the graph, it can be inferred that the validation error of S-DC-CNN trained using ($L_{imit}$ and $L_{rec}^S$) and ($L_{AT}$ and $L_{rec}^S$) is lower than training the network with $L_{rec}^S$. This shows the contribution of both $L_{AT}$, $L_{imit}$ terms in producing lower validation error. However, the following things can also be inferred from the graph 1) using $L_{rec}^S$, $L_{imit}$ and $L_{AT}$ provides validation error almost equal to that of $L_{rec}^S$ and $L_{AT}$ and 2) validation error of S-DC-CNN trained using $L_{AT}$ and $L_{rec}^S$ is lower than training the network with $L_{imit}$ and $L_{rec}^S$. This demonstrates that attention transfer is a key tenet of effective knowledge distillation.

## 5. Conclusion

We proposed a knowledge distillation (KD) framework for image to image problems in the MRI workflow in order to develop compact, low-parameter models without a significant drop in performance. We propose obtaining teacher supervision through a combination of attention transfer and imitation loss. We demonstrated its efficacy on the DC-CNN network and show consistent improvements in student reconstruction across datasets and acceleration factors.

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
