# OpenReview forum: "KD-MRI: A knowledge distillation framework for image reconstruction and image restoration in MRI workflow"
_MIDL.io/2020/Conference — MIDL 2020_

### Official Review · AnonReviewer3 · 2020-02-26
**They authors introduced the method of achieving knowledge distillation on two previous networks, and applied the method for medical image reconstruction and restoration.**

**Rating:** 3
**Confidence:** 4
**Recommendation:** Poster

**Summary:**

The paper shows the method of achieving knowledge distillation on two previous networks, i.e., DC-CNN and VDSR. Its experiments show that the proposed method delivers state-of-the-art performance on MRI reconstruction and super-resolution. The knowledge distillation method is significant in decreasing the cost of computation and storage thanks to its ability of compressing models.

**Strengths:**

The paper provides good validation work regarding to the effectiveness of the proposed framework on different networks and datasets. They showed the performance in MRI reconstruction using cardiac, brain, and knee MRI datasets. They also evaluated the method in MRI super-resolution.

**Weaknesses:**

1. The authors did not provide the performance under different compression rates. In the ablation study, the paper studies the performance under different settings of loss functions, but does not provides the performance under different compression rates. For example, the number of convolutional layers in student DC-CNN is set to be 3, but it could be set to other values from 1 to 4. The performance under high compression rates is more powerful to verify the ability of compressing models.

2. The presentation of model complexity is weak. The significance of knowledge distillation methods is model compression, so the number of parameters and the run time of compared models should be clearly presented.


**Detailed Comments:**

The performance of the knowledge distillation framework on other image restoration tasks should be provided, since “image restoration” is included in the title.

**Justification Of Rating:**

The knowledge distillation method is useful in compressing models. The methodological novelty is limited. The method has been applied to several medical image reconstruction, but the restoration validation work is limited.

**Paper Type:**

validation/application paper

**Questions To Address In The Rebuttal:**

1. Please provide the performance under different compression rates.

2. Please present the model complexity clearly.


**Special Issue:**

no

---

> ### Author Response · Authors · 2020-03-25
> **Compression rate performance, Model complexity, Image restoration**
>
> We thank the reviewer for their comments.
>
> 1. Please provide the performance under different compression rates.
>
> Please find the performance for different compression rates. D5C3 is reported in paper, while D5C4 and D5C2 are obtained for rebuttal. All these networks are trained with the assistance of D5C5 (Teacher).
>
> D5C5 ( Teacher )
> HFN = 0.63 +/- 0.16
> PSNR = 28.45 +/- 3.13
> SSIM = 0.8335 +/- 0.06
>
> D5C4 (kd)
> HFN = 0.63 +/- 0.16
> PSNR = 28.45 +/- 3.16
> SSIM = 0.8334 +/- 0.06
>
> D5C3(kd)
> HFN = 0.65 +/- 0.15
> PSNR = 28.11 +/- 3.17
> SSIM = 0.8236 +/- 0.07
>
> D5C2(kd)
> HFN = 0.71+/- 0.13
> PSNR = 27.03 +/- 3.20
> SSIM = 0.7894 +/- 0.07
>
> 2. Please present the model complexity clearly.
>
> Section 4.3.2 (Parameter count and running time) has been devoted to understanding the model complexity. We have shown that: 1) S-KD-DC-CNN gives 65% parameter reduction as compared to T-DC-CNN, 2) The CPU running time of S-KD-DC-CNN  is nearly 2 times lower than that of T-DC-CNN, 3) The GPU running time of S-KD-DC-CNN  is nearly 1.5 times lower than that of T-DC-CNN.
>
>
> 3. The performance of the knowledge distillation framework on other image restoration tasks should be provided, since “image restoration” is included in the title.
>
> As mentioned in the Introduction, “We also extend our framework for MRI super-resolution and obtained encouraging results which are presented in Appendix A”. In Appendix A, we have adapted the VDSR architecture to our proposed framework and have shown promising results which are tabulated in Table 2.

---

### Official Review · AnonReviewer2 · 2020-03-13
**Readable paper and interesting method**

**Rating:** 3
**Confidence:** 4
**Recommendation:** Poster

**Summary:**

This paper proposed knowledge distillation (KD) framework for memory-efficient models without significant drop of performance. The authors also introduced attention-based feature distillation and imitation loss. Experimental results for MRI reconstruction and super-resolution demonstrate that the proposed KD framework provides significant improvement over the method without KD.

**Strengths:**

The authors applied novel knowledge distillation framework to the MRI reconstruction. To better learn the intermediate representation from teacher, attention-based feature distillation was proposed. Imitation loss function is further proposed for a regularizer.

**Weaknesses:**

- Although the concept of knowledge distillation is very interesting, one limitation is that student network is highly dependent on the teacher network. If the performance of teacher network is poor, how does the student network overcome the teacher?
- Lack of details on dataset: How is the images cropped to 150x150 for ACDC dataset?
- Image resolution is poor, for example, figure 3. Please use more high resolution without bounding box.


**Justification Of Rating:**

The paper is well written and easy to follow. The authors explained the proposed methods and demonstrated the effectiveness with extensive experiments. Overall, this is an interesting and solid approach.

**Paper Type:**

methodological development

**Special Issue:**

yes

---

> ### Author Response · Authors · 2020-03-25
> **DC-CNN ( SOTA ) is used as teacher to provide assistance.**
>
> We thank the reviewer for their comments.
>
> According to knowledge distillation, the objective of the teacher network is to assist and improve the performance of the student network. In our context, the reviewer’s concern of the teacher network performing poorly does not hold, since we use the state-of-the-art DC-CNN architecture as our teacher network. But for a general scenario, the reviewer's concern is prudent and requires research along the lines of training the teacher along with the student so as to improve the robustness of the model. Also, our primary motive in this paper is to not overcome the teacher’s performance but to only improve the reconstruction quality of the student network and bridge the performance gap between teacher and student.
>
> Automated Cardiac Diagnosis Challenge (ACDC) consists of 150 and 50 patient records for training and validation respectively. We extracted the 2D slices and center cropped to 150×150 because of varying dimensions.

---

### Official Review · AnonReviewer1 · 2020-03-15
**Weak accept**

**Rating:** 3
**Confidence:** 4
**Recommendation:** Poster

**Summary:**

In this work, is proposed a knowledge distillation framework for MRI workflows in order to develop
compact, low-parameter models without a significant drop in performance. A combination of the attention-based feature distillation and imitation loss is used to demonstrate the effectiveness of this application on the MRI reconstruction architecture, DC-CNN.

Extensive experiments using Cardiac, Brain, and Knee MRI datasets for 4x, 5x and
8x accelerations are made. For the Knee dataset, the student network achieves 65% parameters reduction, 2x faster CPU running time, and 1.5x faster GPU running time with a limited drop in the accuracy.

The proposed solution is compared with other feature distillation methods, includung  an ablative study to understand the significance of the different components.

**Strengths:**

Adequate comparison with the state-of-the-art methods is proposed in the experimental section.

The proposed method obtains the lower error in comparison to other methods when parameters of the model are reduced.

An ablation-study is also made to show the improvement of different components of the pipeline.


**Weaknesses:**

-The proposed pipeline is a combination of existing approaches [(Saputra et al., 2019)+(Komodakis and Zagoruyko, 2017)]

-I am not very convinced that in a medical application it is acceptable to drop accuracy to save some computational cost and model memory.  However, parameters reduction can be useful also for other aspects such as to obtain a better generalization of the model. I would suggest the authors to make their motivations for the proposed application stronger.

**Detailed Comments:**

Figure 2 should report a line on top of the figure to make more clear what each column represent.



**Justification Of Rating:**

The proposed approach shows promising results. An adequate experimental section highlights the obtained improvements with respect to the other methods, and an ablation study to understand the contribution of each component.

The motivations of using the proposed approach in a medical context to allow drop in accuracy to save computational cost and memory are not convincing.

**Paper Type:**

methodological development

**Questions To Address In The Rebuttal:**

The authors should make the motivations of using their pipeline in a medical application stronger.






**Special Issue:**

no

---

> ### Author Response · Authors · 2020-03-25
> **Primary focus is to achieve parameter reduction without significant drop in performance.**
>
> We thank the reviewer for their comments.
>
> We agree with the reviewer on the statement: “I am not very convinced that in a medical application it is acceptable to drop accuracy to save some computational cost and model memory“. However, since this is an initial foray into the applications of KD-based approach for the MRI workflow, our primary focus is to show that the parameter reduction can be achieved without a significant drop in performance. We believe that this performance gap between the Teacher and Student can be bridged by recent advancements in model compression focusing on classification tasks where competitive performances have been achieved.

---

### Official Review · AnonReviewer4 · 2020-03-17
**Interesting use of knowledge distillation framework for image reconstruction in MRI, shows only slight improvements compared to zero-filling though**

**Rating:** 3
**Confidence:** 4
**Recommendation:** Poster

**Summary:**

key ideas:
In this article, the authors tackle the clinically important problem of reconstruction of accelerated MRI. They use the concept of knowledge distillation to reduce the number of parameters and consequently running time for reconstruction of undersampled MRI data, which is an ill-posed problem. They also compare different ways of knowledge distillation.

experiments:
The authors examine performance of their proposed algorithms and other earlier algorithms on three different datasets - brain, cardiac and knee MRI - with different subsampling factors (4x,5,8x). While the experiments themselves are set up nicely, the results are a bit disappointing.  In the qualitative results the differences are very small and almost invisible. Regarding the quantitative analysis, the evaluation metrics used (Peak Signal-to-Noise Ratio (PSNR) and Structural Similarity Index (SSIM) metrics ) are completely appropriate for the task, but there is largely no difference between the proposed method that employs KD and a regular student network. Also the authors state they will use the Wilcoxon signed rank test with an alpha of 0.05 to assess statistical significance, but this is never done.

significance:
The clinical problem of that the article addresses is very relevant, but the experiments are not convincing enough for me that the knowledge distillation really gives enough of a performance boost.

**Strengths:**

The paper tackles an important problem in MRI. Reconstruction of undersampled k-space data is a challenging problem and there have been a lot of attempts in solving it. The paper adds to the previous approaches by trying to reduce the number of parameters fitted and computational time with a knowledge distillation approach.




**Weaknesses:**

While the approach is interesting and the experiments seem to be performed fine, the results are a bit disappointing.  In the qualitative results the differences are very small and almost invisible. Regarding the quantitative analysis, the evaluation metrics used (Peak Signal-to-Noise Ratio (PSNR) and Structural Similarity Index (SSIM) metrics ) are completely appropriate for the task, but there is largely no difference between the proposed method that employs KD and a regular student network. Even the performance of a simple zero-filling is not that far off. This should be clearly addressed in the discussion.

Also the authors state they will use the Wilcoxon signed rank test with an alpha of 0.05 to assess statistical significance, but this is never done.

The submitted pdf was way over the recommended 8 page limit- the actual paper ran onto the 10th page and with appendixes the pdf had 18 pages. Important things, such as the quantitative results were hidden in the appendix which is not acceptable.

**Justification Of Rating:**

The paper is generally nicely written and addresses an important problem. The authors have also a nice idea for improving the current solutions on the issue of undersampled MRI reconstruction. The setup of the experiments is nice, but the results disappointing. Also the actual presentation of the results and discussion is subpar.

**Paper Type:**

both

**Questions To Address In The Rebuttal:**

- Please make the quantitative results (Table 2)  a part of the main manuscript as they are quite essential.

- As stated in the paper please use the Wilcoxon signed rank test with an alpha of 0.05 to assess statistical significance. Then address the outcome which will (after looking at Table 2) probably show no significant improvements in PSNR and SSIM for your proposed KD network compared to a regular student network and only a small improvement compared to zero filling.

- Length of the paper - currently the paper is much over the recommended 8 page limit, try and cut down the background sections and add a thorough discussion on how much the performance is actually boosted.

**Special Issue:**

no

---

> ### Author Response · Authors · 2020-03-25
> **Qualitative and quantitative results are significant. Better recovery of structures. p < 0.05 in Wilcoxon signed rank test**
>
> We thank the reviewer for their comments.
>
> In our work, we have proposed a knowledge distillation framework which can work on different stages of MRI workflow, like image reconstruction and restoration.
>
> In our main paper, we primarily focused on MRI reconstruction to showcase our framework’s capability. We chose DC-CNN for our experiments. The qualitative and quantitative results are presented in Figure 2 and Table 1 respectively. In Figure 2, we have clearly illustrated the qualitative differences between the teacher, student and our proposed KD-based approach. It is evident from the figure that our model gives a better reconstruction than the student (structures are smudged in the student model), and approaches the reconstruction quality of the teacher.
> We also conducted a statistical significance test (Wilcoxon signed rank (alpha < 0.05)) and found that the performance improvement was significant in relation to the student network. Also, to the best of our knowledge on MRI reconstruction, the quantitative gains shown in PSNR and SSIM metrics are substantial.
>
> In the supplementary section, we validated our proposed framework’s adaptability to restoration problems by testing it for MRI super-resolution, and thereby showcasing its potential to be used across the MRI workflow. The state-of-the-art architecture VDSR is chosen as our MRI super-resolution network. Quantitative and Qualitative comparisons could be found in Figure 5 and Table 2 respectively. Similar to MRI reconstruction, we have shown that our model recovers finer structures unlike the student network. Also, the performance improvement shown by our model was found to be statistically significant (Wilcoxon signed rank (alpha < 0.05)) in relation to the student network.
>
> To address the reviewer’s concerns about zero-filled results in Table 2: The quality of zero-filled depends on several factors like dataset type, type of degradation, amount of data, scale factor etc. For our experiments, we chose the brain MRI dataset, and the low resolution images (4x SR) are obtained by only considering the low frequency components in the centre (square region) while filling the remaining components with zeros. The teacher network (VDSR, 11 layers) has a gain of 2db over zero-filled, while our model (VDSR, 7 layers + KD) also has a gain of around 2db. In MRI super-resolution, gains over 1 db are in general deemed to be substantial.

---

### Meta-Review · Area_Chair1 · 2020-04-06
**MetaReview of Paper9 by AreaChair1**

**Rating:** 3
**Recommendation For Accepted Papers:** Poster

**Metareview:**

The paper presents a method for MRI reconstruction of undersampled k-space using a knowledge distillation approach.

According to the 3 reviewers, the paper is well written although too long for a MIDL paper.

Method novelty is limited as it seems a combination of existing approaches.

From an applicability point of view, the reviewers and I are concern about the drop in quality for such a modest speed-up factor.




**Paper Type:**

methodological development

**Special Issue:**

no

---

### Decision · Program_Chairs · 2020-04-11

Accept